# Using Physics Knowledge for Learning Rigid-body Forward Dynamics with Gaussian Process Force Priors

**Lucas Rath**[*,1]     **A. René Geist**[*,1]     **Sebastian Trimpe**[1,2]

[1]Max Planck Institute for Intelligent Systems, Stuttgart
{rath, geist}@is.mpg.de

[2]Institute for Data Science in Mechanical Engineering, RWTH Aachen University
trimpe@dsme.rwth-aachen.de

**Abstract:** If a robot's dynamics are difficult to model solely through analytical mechanics, it is an attractive option to directly learn it from data. Yet, solely data-driven approaches require considerable amounts of data for training and do not extrapolate well to unseen regions of the system's state space. In this work, we emphasize that when a robot's links are sufficiently rigid, many analytical functions such as kinematics, inertia functions, and surface constraints encode informative prior knowledge on its dynamics. To this effect, we propose a framework for learning probabilistic forward dynamics that combines physics knowledge with Gaussian processes utilizing automatic differentiation with GPU acceleration. Compared to solely data-driven modeling, the model's data efficiency improves while the model also respects physical constraints. We illustrate the proposed structured model on a seven joint robot arm in PyBullet. Our implementation of the proposed framework can be found here: https://git.io/JP4Fs

**Keywords:** Machine learning, Robotics, Analytical mechanics

## 1 Introduction

An accurate dynamics model is essential for computing a robot's actions via model-based control approaches such as model predictive control [1–4] or model-based reinforcement learning [5–10]. Yet, in many robotic systems including legged robots and robot arms, physical phenomena such as friction and elasticities considerably affect the dynamics, are often environmental dependent, and unknown a-priori which in turn aggravates analytical physics modeling. As an alternative to analytical modeling, recent works resort to learning dynamics solely from input-output data using

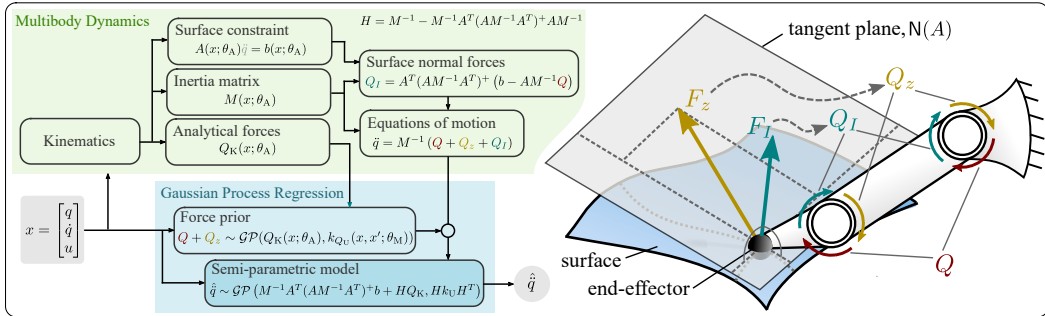

Figure 1: Computational graph of the proposed framework. A robot arm's forward dynamics model is augmented by a GP approximating unknown forces $Q_{\mathrm{U}}$. The semi-parametric model predicts acceleration $\ddot{q}$ while potentially including knowledge in form of analytical functions $\{M, A, b, Q_{\mathrm{K}}\}$.

---

[*]equal contribution.

5th Conference on Robot Learning (CoRL 2021), London, UK.

regression models. Yet, data collection on robotic systems must be done in real-time using potentially sub-optimal control strategies that only operate in small regions of the system's state-space. Therefore, data sets that stem from physical systems are often limited in size and information value. This raises a considerable obstacle for successfully training data-driven models such as a neural network (NN) or a Gaussian process (GP), as these are data-hungry compared to physics models and usually do not extrapolate well. To overcome the limitations of data-driven modeling, recent works combine analytical mechanics with data-driven models resulting in so-called structured models or graybox models. In structured models, the target function, that is the parts of an analytical physics model that are difficult to model a-priori, is being approximated by a carefully designed data-driven model. By use of physical prior knowledge, the data efficiency of a structured model is potentially improved by: i) Reducing the *complexity* of the target function; ii) Reducing the *dimensionality* of the target function; or iii) Building prior knowledge on *physical properties* of the target function into the data-driven model. This target function ideally contains all erroneous parts of the physics model while the sample-complexity of a regression model is smaller when learning such a target function compared to simply learning the residual function in the physics model's output.

In general, finding a suitable candidate for a target function inside an analytical physics model is not trivial. In the following, we suggest that for sufficiently rigid bodies the majority of the analytical model errors arise from an inaccurate depiction of forces. As many of these forces are potentially uncertain and stochastic in nature, we propose a probabilistic structured dynamics model in which we approximate unknown forces inside an analytical dynamics model by a GP. In particular, the contribution of this work are as follows:

- We propose a modelling framework using PyTorch that combines a library for (implicitly constrained) rigid-body mechanics with a novel implementation of structured multi-task GP regression. We analyze the proposed framework by learning the dynamics of a seven dimensional robot arm whose end-effector touches a surface in PyBullet.

- We show that analytical parameters can be efficiently estimated alongside a seven-dimensional structured multi-output GP with 154 hyperparameters.

- We emphasize the new research opportunities that evolve from the proposed framework. Such as the prediction of implicit contact forces by solely resorting to measurements of the robot's constrained acceleration as well as using Baumgarte stabilization to enforce that long-term trajectory predictions made using the structured GP respect implicit constraints.

### 1.1  Problem Formulation

We assume that the following assumptions apply to a robot whose forward dynamics (FD) we aim to identify:

**Assumption 1** *The underlying kinematics function of the robot arm and kinematic surface equation, as well as their parameters, are known a-priori.*

**Assumption 2** *The bodies (aka links) of the mechanical system are rigid.*

Further, we assume that the state of a robot's forward dynamics description is given by $q \in \mathbb{R}^{n_q}$, $\dot{q}$, $\ddot{q}$, and $u \in \mathbb{R}^{n_u}$ being the joint angles, velocity, acceleration, and control input respectively. As discussed in Section 1.3, these assumptions are commonly found in literature on nonlinear robot dynamics identification [11–15]. Given input points $x_k = [q_k, \dot{q}_k, u_k]^\mathsf{T}$, acceleration measurements are modeled as

$$y_k(x_k) = \ddot{q}(x_k) + \epsilon_k \ \text{ with } \ \epsilon_k \sim \mathcal{N}\left(0, \Sigma_y\right), \tag{1}$$

where $\Sigma_y$ denotes a diagonal matrix containing the measurement variances. Further, an informative data-set $\mathcal{D} = \{X, Y\}$ is available consisting of the input data vector $X = [x_1^\mathsf{T} \ldots x_N^\mathsf{T}]^\mathsf{T}$ and the output data vector $Y = [y_1^\mathsf{T} \ldots y_N^\mathsf{T}]^\mathsf{T}$. The main objective of this work is, for a high-dimensional robot arm, to propose a modeling framework that enables the data-efficient prediction of $\mu_{X^\star}^{\ddot{q}} | \mathcal{D} = \mathbb{E}(\ddot{q}(\boldsymbol{x}_*) | \mathcal{D})$ and $K_{X^\star, X^\star}^{\ddot{q}} | \mathcal{D} = \mathrm{cov}(\ddot{q}(\boldsymbol{x}_*) | \mathcal{D})$ at a test input $\boldsymbol{x}_*$ with a structured GP model $\hat{\ddot{q}} \sim \mathcal{GP}(m_{\ddot{q}}, K_{\ddot{q}})$. We refer to $\hat{\ddot{q}}$ as structured as this model results from combining physical prior knowledge underlying the system's equations of motion (EOM) with a GP prior placed on the system's unknown forces $\hat{Q}_\mathrm{U} \sim \mathcal{GP}(0, K_\mathrm{U})$.

## 1.2 Forward Dynamics of Robot Arms

In what follows, we omit the arguments of functions to avoid cluttered notation if these are clear from the context. In this work, we limit the discussion to Lagrangian FD of robot arms as described by equations of motion (EOM) taking the form

$$\ddot{q} = f_{\mathrm{A}}(x; \theta_{\mathrm{A}}) = M^{-1}\left(Q + Q_z + Q_I\right), \tag{2}$$

with the analytical model parameters $\theta_{\mathrm{A}}$, symmetric and positive definite inertia matrix $M(q; \theta_{\mathrm{A}})$, torques acting on the unconstrained system $Q(x; \theta_{\mathrm{A}})$, torques $Q_z(x; \theta_{\mathrm{A}})$ caused by Cartesian end-effector friction forces $F_z$, and torques $Q_I(x; \theta_{\mathrm{A}})$ that are caused by Cartesian surface normal forces $F_I$ also acting on the end-effector. That is, if $Q$ presses the robot's end-effector onto a surface, the surface applies as a reaction the force $F_I$ that always remains normal to the surface. The transformation of $F_I$ into generalized coordinates yields the torques $Q_I$. The force $F_I$ often causes dissipative forces between the end-effector and the surface. These dissipative forces $Q_z$ lie always tangential to the surface. The torques acting in the unconstrained system $Q(x; \theta_{\mathrm{A}}) = Q_D + Q_u + Q_G + Q_C$ consist of bias forces $Q_C$, gravitational forces $Q_G$, dissipative forces inside the joints $Q_D$, and actuation forces $Q_u$. Often in literature $Q_u + Q_D$ is denoted by $\tau$ being referred to as joint torques which are the torques that are measurable in a robot's joints. The analytical model parameters $\theta_{\mathrm{A}}$ can be divided into kinematic parameters (*e.g.*, the length between adjacent joints), and inertia parameters (*e.g.*, CoG positions, masses, and inertias). The a priori estimated model parameters $\theta_{\mathrm{A}}$ often deviate from the real physical values of the system and need to be further identified using data. For kinematic trees, such as most robot arms, the FD $f_{\mathrm{A}}(x; \theta_{\mathrm{A}})$ are straightforwardly obtained using the articulated body algorithm. Alternatively, inverse dynamics (ID) seek a mapping from $\{q, \dot{q}, \ddot{q}\}$ to $u$. However, unlike ID, FD allow the direct simulation of a system's motion. In turn, one could use an accurate structured FD model to train an RL policy in simulation. Moreover, the largest noise usually lies on acceleration estimates denoting the output of FD models. In the case of GPs it is simpler to model noise on the outputs than inputs [16].

In the absence of end-effector forces, the coordinates $q$ uniquely define every admissible state of the robot arm such that the robot's acceleration is given as $\ddot{q} = M^{-1}Q$. However, the presence of $Q_I$ enforces that the end-effector's motion during contact respects the holonomic surface constraints $c(q) = 0$, with $c(q) : \mathbb{R}^{n_q} \to \mathbb{R}^m$, effectively reducing the set of admissible states the robot arm may acquire. Instead of working directly with $c(q) = 0$, one can compute its second time-derivative to obtain a constraint equation that is linear in $\ddot{q}$, writing

$$A\ddot{q} = b, \tag{3}$$

with $A(q, \dot{q}; \theta_{\mathrm{A}}) : \mathbb{R}^{2n_q} \to \mathbb{R}^{m \times n_q}$, $b(q, \dot{q}; \theta_{\mathrm{A}}) : \mathbb{R}^{2n_q} \to \mathbb{R}^m$, and $m < n_q$. As detailed in [17], the part of all forces doing virtual work, $Q' \in Q$, and the implicit constraint forces $Q_I$ doing no virtual work, must lie in $Q_z \in Q' \in \mathsf{N}(A)$ as well as $Q_I \in \mathsf{R}(A^{\mathsf{T}})$, with the null space of $A \in \mathbb{R}^{m \times n_q}$ being defined as $\mathsf{N}(A) = \{\ddot{q} \in \mathbb{R}^{n_q} : A\ddot{q} = 0\}$ and its range space as $\mathsf{R}(A) = \{b \in \mathbb{R}^m : \exists \ddot{q} \in \mathbb{R}^{n_q} \text{ such that } b = A\ddot{q}\}$ [18]. The fact that $Q_I \in \mathsf{R}(A^{\mathsf{T}})$ motivates its parametrization in terms of Lagrange multipliers $\lambda$, writing $Q_I = A^{\mathsf{T}}\lambda$. As detailed in [19] one obtains an explicit formula for the Lagrange multipliers as $\lambda = (AM^{-1}A^{\mathsf{T}})^+ \left(b - AM^{-1}Q\right)$, where the $A^+$ denotes the Moore-Penrose (MP) pseudo inverse of $A$. Subsequently, the system's EOM are obtained as

$$\ddot{q} = M^{-1}Q_b + H\bar{Q}, \tag{4}$$

with $H = M^{-1} - M^{-1}A^{\mathsf{T}}(AM^{-1}A^{\mathsf{T}})^+AM^{-1}$, $\bar{Q} = Q + Q_z$, and $Q_b = A^{\mathsf{T}}(AM^{-1}A^{\mathsf{T}})^+b$. Further details on implicitly constrained dynamics are given in Section 1 of the supplementary material.

## 1.3 Related Work

The identification of the errors in a robot's analytical EOM often forms a metaphorical Gordian knot consisting of errors that arise from the interplay of wrong analytical parameters (*e.g.*, inertia parameters), wrong analytical functions (*e.g.*, friction forces), and in the worst case the analytical model class itself being an inaccurate depiction of the real dynamics (*e.g.*, describing flexible bodies with rigid-body dynamics). In what follows, we discuss recent works that combine data-driven models with rigid-body dynamics.

**Parameter estimation of rigid-body dynamics models**  Pioneering works on robot dynamics identification focused on the estimation of analytical parameters. For example, the well-known approach of Atkeson et al. [11] uses the linearity of a rigid robot arm's ID with respect to its parameters to estimate these via linear regression. With the increased popularity of libraries for automatic differentiation, recent works [12–15] use gradient-based optimization to estimate the physical parameters inside the analytical model from data. Such approaches work well if the robot arm is rigid and the forces acting on it are sufficiently known. However, the parameter estimation of an insufficient analytical representation of a robot's dynamics leads to physically inconsistent parameter estimates as well as unsatisfactory model accuracy. While we assume that the analytical model differs from the real dynamics, we also implement a differentiable analytical model in PyTorch such that we can estimate analytical parameters *alongside* the parameters of a data-driven model.

**Building rigid-body dynamics into neural networks**  As an alternative approach to analytical modeling, [20–23] model dynamics via the Euler-Lagrange equations in which a NN approximates the system's Lagrangian or the entries of the inertia matrix $M$. In turn, such a Lagrangian NN increases the flexibility of the functional approximation of mass-related dynamic terms $\{M, Q_G, Q_C\}$ compared to analytical models while still respecting energy conservation. These works often assume $Q_u \hat{=} \tau$ and thereby potentially neglect the joint friction $Q_D$. If forces are the major sources of analytical errors and mass related quantities are not, then Lagrangian NNs add flexibility through the NN in the functional representation of a robot's dynamics where it might not be required. So far this hypothesis has not been refuted, as the works on Lagrangian NNs test their algorithms solely on pendulums or low-dimensional robot arms without end-effector contacts.

A recent branch in structured modeling combines analytical models of mass-related quantities and presumably known forces with data-driven modeling. Lutter et al. [24] augments an analytical FD model with a NN approximating $\tau$. The parameters of the analytical model and NN are estimated jointly via automatic differentiation. Recently, [10] and subsequently [25] combined an analytical simulation with a pre-trained NN – predicting measured joint torques $\tau$. Notably, $Q_I$ and $Q_z$ are modeled analytically as detailed in [9] while their physical parameters are varied to robustify the trained control policy towards parameter uncertainties. Significantly, NN control policies that were trained on such structured models achieved so far unseen robustness of a quadruped's locomotion policy. These seminal works indicate that the gap between modeled and real dynamics (aka sim2real gap) can be closed by consciously combining analytical mechanics with data-driven modeling.

**Building rigid-body dynamics into Gaussian Processes**  On real robotic system's, noise and uncertainty are often significantly impacting the robot's dynamics. For example, a friction force denotes a macroscopic abstraction of microscopic tribologic phenomena which aggravates deterministic modeling. In addition, depending on the robot's sensors, the changes in the joint torques due to elasticities and backlash are often not observed accurately and subsequently also introduce uncertainty into the dynamics. As shown in [6, 26], the synthesis of a robot's control policy can significantly benefit from the availability of an uncertainty measure for the planned motion. Popular models for the identification of uncertain and noisy dynamics are either building on Bayesian linear regression as *e.g.*, [27], or often resort to GPs [8, 28–32]. GPs allow for the incorporation of various model assumptions through the covariance function (kernel) and are often more data-efficient compared to NNs. Yet, vanilla GP regression requires the computation of the inverse covariance matrix which demands a computational complexity of $\mathcal{O}((DN)^3)$ and a memory requirement of $\mathcal{O}((DN)^2)$ where $D$ denotes the number of the GP's correlated outputs. In addition, the inversion of the covariance matrix is prone to numerical problems while the computational efficient implementation of multi-output GPs forms a considerable obstacle. Despite these challenges, several models have been proposed that combine GP regression with analytical dynamics.

Nguyen-Tuong and Peters [33] used [11] as a linear parametric mean of a GP model (cf. [16, Chapter 2.7]). If Assumption 2 applies onto the robot arm, the GP in [33] effectively approximates the joint torques $\tau$ (in the absence of end-effector forces). As outlined in Section 2, we also approximate unknown forces through a GP. Albeit, as a robot arm's FD are rarely linear with respect to analytical parameters, structured GP regression is more challenging for FD compared to ID. Conceptually being similar to [33], Saveriano et al. [34] approximate the residuals of analytical transition dynamics, being the mapping from $\{q_k, \dot{q}_k, u_k\}$ to $\{q_{k+1}, \dot{q}_{k+1}\}$, via one-dimensional GPs. Our approach does not directly approximate dynamics residuals with GPs, but instead approximates the residual function by placing a GP prior *inside* the analytical model.

Cheng et al. [35] placed a GP on the Lagrangian inside a robot arm's Euler-Lagrangian ID. Yet, for GPs it is not clear how to formulate such a model for FD and to which extent the end-effector forces can be incorporated. A Lagrangian GP leverages that the derivative of a GP if it exists, is itself a GP [36]. Jidling et al. [37] emphasized that more generally GPs are closed under linear functionals. In turn, the authors proposed a linearly transformed GP such that its predictions fulfil a linear operator equation *e.g.*, the mechanical stress field inside a linear elastic material. Our work also transforms a GP by a linear operator, that is, the matrix-valued function $H$ and in turn enforce that predictions respect the linear-affine equation (3). Methodically, our work builds on Geist and Trimpe [38]. In addition to [38], we suggest placing a GP on the unknown *forces* of the system rather than its unconstrained *acceleration*. As we detail in Section 2.1, this alternative approach potentially improves the model's sample-efficiency as well as eases the inclusion of prior knowledge on force properties into the GP. In addition, [38] only tested their model on low-dimensional systems such as a point-mass sliding over a surface. In contrast, we propose a framework that allows for learning high-dimensional dynamics. Our framework is significantly faster compared to [38] by resorting to automatic differentiation with GPU accelerated computation of gradients, and shows how to transform regression models via recursively computed rigid-body dynamics.

## 2 Semi-parametric Regression of Dynamics with Gaussian Processes

In this section, we extend the findings of Geist and Trimpe [38] and use knowledge on the parametric functions $\{M, A, b\}$ as stated in (2) and (3) to linearly transform a GP. Here, we leverage that analytical mechanics defines the acceleration of a system in terms of affine transformations of force functions which we approximate by a GP. An introduction to multi-output GP regression is provided in Section 2 of the supplementary material.

### 2.1 Combining Data-driven with Analytical Modeling

In this section, we propose a framework that combines Gaussian process regression with parametric analytical modeling. To develop a thorough understanding of the underlying assumptions we first discuss the main building blocks of the analytical EOM (4) as also shown schematically in Figure 1. As the dynamics (4) are expressed in generalized coordinates, the entries of $M$, $Q_G$, $Q_C$, $A$, and $b$ depend on *kinematic functions*. The kinematic functions themselves depend on the system's state $\{q, \dot{q}\}$ and kinematic parameters such as the length of the bodies. In practice, precise estimates for the kinematic parameters are obtained from computer-aided design (CAD) or kinematic parameter estimation techniques. In Cartesian space, an accurate model for (3) that is $\{A, b\}$ can be straightforwardly obtained by *e.g.*, using a camera system. In turn, (3) is obtained in generalized coordinates using kinematic coordinate transformations. Besides kinematic expressions, the *inertia matrix $M$* contains inertia parameters, namely the CoG positions, inertias, and masses. We assume that a prior for the inertia parameters can be obtained using also CAD. Yet, for FD it is critical that we mitigate errors in the inertia parameters being part of $\theta_A$ as all forces in (4) are multiplied by $M^{-1}(x; \theta_A)$.

As pointed out in Section 1, it is often difficult to analytically describe force functions that arise from elasticities and friction phenomena. Therefore, we divide the analytical forces $\bar{Q}$ into a known part $Q_K(x; \theta_A)$ and an unknown part $Q_U(x)$. The term *known* is defined in the following as the existence of an optimal $\theta_A$, $\theta_A^\star$, such that the error between a physical function and its analytical description in terms of $\theta_A$ approaches zero. Knowledge on $\theta_A^\star$ can be unavailable a-priori such that its entries need to be estimated from data. Subsequently, we model the generalized forces as a GP, writing

$$\hat{\bar{Q}} \sim \mathcal{GP}(Q_K(x; \theta_A), \, k_{Q_U}(x, x'; \theta_M)), \tag{5}$$

in which $k_{Q_U}$ denotes an appropriately chosen kernel function and $\theta_M$ its hyper-parameters. By inserting (5) into (4), we obtain a structured model for the system's FD as

$$\hat{\ddot{q}} \sim \mathcal{GP}(M^{-1}Q_b + HQ_K, \, Hk_{Q_U}H^\mathsf{T}). \tag{6}$$

The above model is itself a GP as $\ddot{q}$ follows from an affine transformation of $\bar{Q}$. To make predictions with (6), the GP conditional posterior formula is used. Note that in (6), we placed the GP prior on the unknown forces instead on placing it on the unconstrained acceleration $M^{-1}Q_U$ as proposed by [38]. Since $M$ is known and its entries are usually nonlinear functions, learning a function after it has been transformed by $M$ is likely to require a more expressive data-driven model to achieve a

comparable prediction accuracy [17]. In addition, as elaborated in Section 3 of the supplementary material, a GP force prior can be designed as a sum of GPs which each model a specific force. In turn, our framework (6) enables a straightforward inclusion of prior knowledge on the properties of forces into the GP. Moreover, a trained GP force prior (5) can provide predictions on other analytical processes such as the generalized normal forces $Q_I$.

## 2.2 Type-II MLE of Structured Gaussian Processes

While a merit of the model presented in (6) forms its simplicity, the optimization of both analytical and kernel parameters poses a significant challenge. In this work, we resort to Type-II maximum likelihood estimation (MLE). MLE forms a standard approach for estimating GP parameters. Given $p(Y|X;\theta) \sim \mathcal{N}(\mu_X^{\ddot{q}}, \Sigma_{X,X}^{\ddot{q}} + \Sigma_Y)$ resulting from (6), type-II MLE seeks a $\theta^\star = \{\theta_A^\star, \theta_M^\star\}$ minimizing the negative log likelihood, writing

$$\theta^\star = \arg\min_{\boldsymbol{\theta}} \frac{1}{2}\left[\left(Y - \mu_X^{\ddot{q}}\right)^\mathsf{T}\left(\Sigma_{X,X}^{\ddot{q}} + \Sigma_Y\right)^{-1}\left(Y - \mu_X^{\ddot{q}}\right) + \log\left|\Sigma_{X,X}^{\ddot{q}} + \Sigma_Y\right|\right] + \frac{N}{2}\log 2\pi. \tag{7}$$

The left term inside the bracket in (7) assigns a cost to the data-fit while the log-determinant term penalises the function's complexity [16, p. 113]. As (6) denotes a multi-output GP such that $D > 1$, the computational cost increases significantly compared to standard GP regression. Moreover, as pointed out in [16], the objective function in (7) is highly nonlinear such that if local optimization techniques are considered we need to restart the optimization several times with random initial parameter values. To solve (7) using restarts for high-dimensional dynamics in reasonable amounts of time, we require a fast computation of analytical parameter gradients. However, such an approach to (7) requires the computation of partial gradients of the analytical parametric function $\mu_X^{\ddot{q}}(x; \theta_A)$ as well as $\Sigma_{X,X}^{\ddot{q}}$ being a transformation of $k_{\ddot{q}}(x, x'; \theta_M)$ by $H(x, \theta_A)$. Fortunately, recent ML libraries such as JAX [39] and PyTorch [40] allow the optimization of functions via automatic differentiation (building on AutoGrad [41]) in combination with libraries for GPU accelerated computation such as XLA. In turn, JAX and PyTorch provide a fast computation of analytical function gradients as long as the function is expressed in terms of an underlying numerical linear algebra library such as Numpy or PyTorch's torch functions. Therefore, we implemented a multi-body library using native PyTorch functions resulting in $\{A, b, M, Q_K\}$ as well as $\{\mu_X^{\ddot{q}}, H\}$. Then, while using the mean and kernel function from GPyTorch [42], we designed a structured GP inference framework from scratch using PyTorch to obtain and optimize the likelihood in (7). The optimization is done using automatic differentiation [43] and GPU acceleration resulting in a significant speed-up compared to numerical gradient approximation techniques.

## 3 Simulation Results

In this section, the proposed structured GP (6) is analyzed and compared to different baseline models. The chosen dynamical system is a KUKA arm with 7 rotational joints simulated in PyBullet [44] as depicted in Figure 2a. A detailed description of the simulation setup is given in Section 5 of the supplementary material. Data is collected by controlling the robot's end-effector along linear trajectories while pressing onto the surface. This task can be seen as an abstraction to a robot arm performing welding, cutting, or marking maneuvers on surfaces. The contact between the end-effector and the surface creates ideal-constraint forces, which avoids the end-effector penetrating the surface, as well as friction forces which are challenging to be modeled beforehand. While running the simulation, we measured the joint angles $q$ and the control inputs $u$. The joint velocity $\dot{q}$ and acceleration $\ddot{q}$ are obtained by using a low-pass filter and appealing to numerical differentiation. After data collection, we applied Farthest Point Sampling [45] to remove adjacent data points as well as reduce the data set's size. The post-processed data is split into a training and a test data set.

**Figure 2b and 3 : Prediction accuracy and data efficiency**   As a baseline for comparison, we trained the parameters of (4) with $Q_z = 0$ as well as a feed-forward NN on ten thousand data points. The trained analytical model and the NN achieved on the test data set a mean absolute error (MAE) of $0.57$ and $0.13$, respectively. For all GP models, we choose a squared exponential (SE) kernel. The 154 hyper-parameters of each GP were estimated according to (7) using Adam

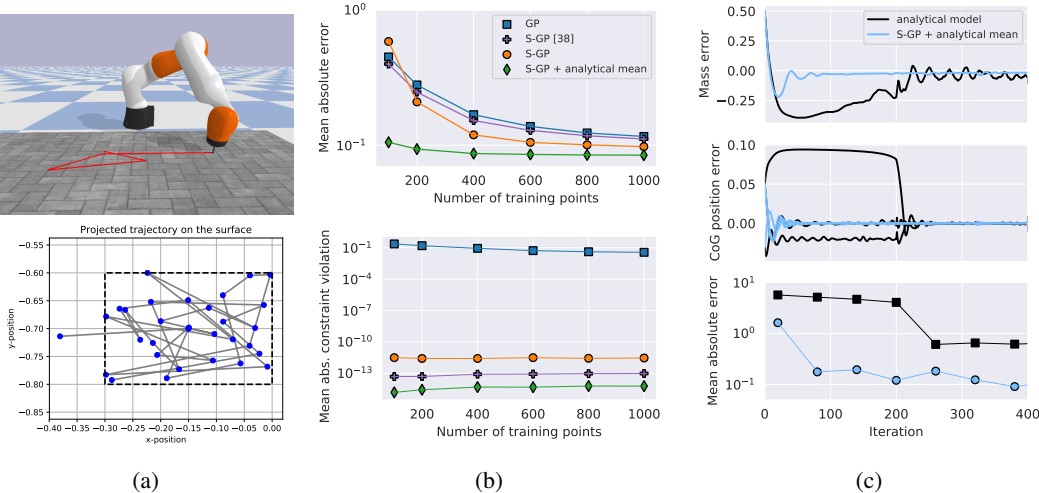

Figure 2: a) Robot arm in PyBullet (Top) and end-effector trajectory during collection of training data (Bottom). b) The hyper-parameters of GP FD models are trained on an increasing number of training points, then the mean absolute acceleration prediction error as well as acceleration-level constraint error on the test data set are compared. c) Parameter optimization results of an analytical model (black) and the proposed structured GP model (blue).

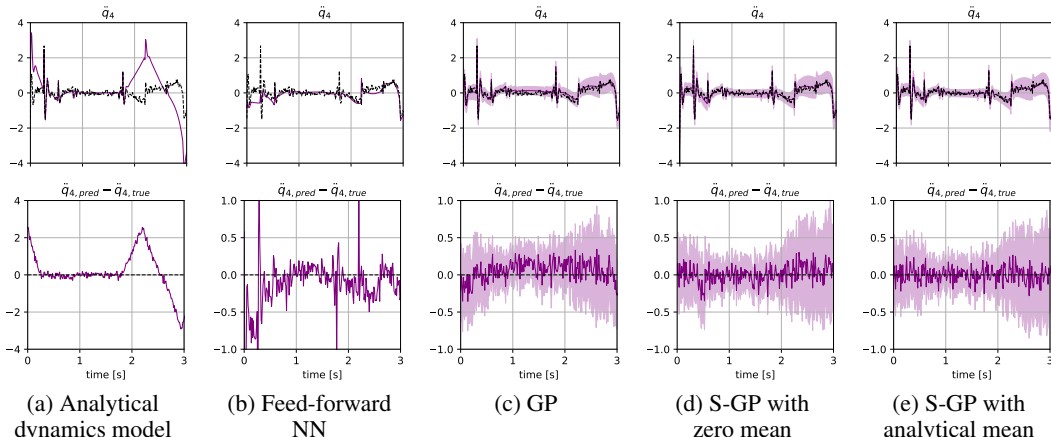

| (a) Analytical dynamics model | (b) Feed-forward NN | (c) GP | (d) S-GP with zero mean | (e) S-GP with analytical mean |

Figure 3: Acceleration predictions $\ddot{q}_{4,\text{pred}}$ (Top) and corresponding prediction error (Bottom) of different forward dynamics models. The black lines depicts the system's noisy acceleration over time, the purple line the predicted mean acceleration and light purple – if available – the $\pm 2$ std. deviation confidence region. Note that the y-axis scaling of Figure 3a deviates from the other figures.

[46] without parameter constraints. Figure 2b illustrates the GPs' test-data MAE of each joint-dimension over an increasing size of the training data set. In this figure, the term "S-GP" refers to the model in (6) with $Q_K = 0$ while the term "S-GP + analytical mean" assumes an analytical model $Q_K = Q_G + Q_C + Q_u$ as the GP's prior mean function. For both S-GP models, we assume that accurate analytical parameters are given, that is $\theta_A = \theta_A^\star$. We assume $Q_G + Q_C$ as known as these analytical functions are being derived solely in terms of known kinematics and inertia functions, as well as the gravitational acceleration constant [17]. The structured GP models are compared to standard GP regression in which each acceleration function is modelled by a single independent SE GP. As shown in Figure 2b (Top), analytical prior knowledge improves the data efficiency of the GP. The proposed S-GP models compares also favorably to placing a GP prior on the system's unconstrained acceleration as initially proposed by [38] being denoted as "S-GP [38]". Moreover, the incorporation of implicit constraint knowledge in (6) significantly reduces the constraint error $A\hat{\ddot{q}} - b$ as illustrated in Figure 2b (Bottom). Figure 3 illustrates different acceleration predictions made with these models using 1000 training points. The analytical baseline model does not contain

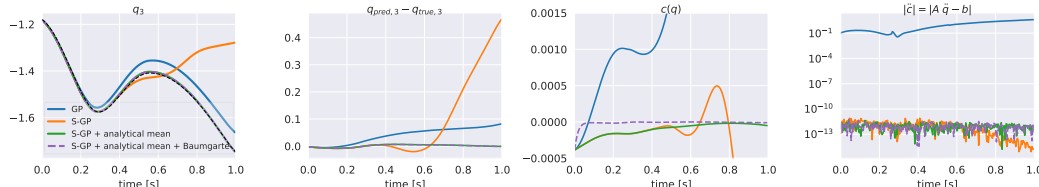

Figure 4: Long-term trajectory prediction and corresponding error functions obtained by numerical integration with integration time step of $(1/240)s$ using the different GP models' acceleration predictions of the fourth output dimension. The true trajectory is depicted by the black dotted line.

a function describing the surface friction which in turn causes the model to yield large prediction errors in the robot arm's joints that are close to the end-effector.

**Figure 2c: Joint estimation of inertia and kernel parameters** Another important aspect of the proposed structured modeling framework forms the simultaneous estimation of $\theta_M$ and $\theta_A$. To illustrate how $\theta_M$ and $\theta_A$ are estimated jointly, we train the same S-GP with $Q_K = 0$ as in the previous simulation on thousand data points. Yet, instead of assuming that we obtained a good prior for $\theta_A$, we now estimate the end-effector's inertia parameters alongside the GP's 154 hyper-parameters. For example, one could imagine that a tool such as a brush or milling machine is being fixed to the end-effector changing its inertia parameters. Figure 2c illustrates the optimization results. Compared to training the parameters of an analytical model without surface friction, the parameter estimates of the proposed S-GP converge faster while the prediction's MAE for all joint-dimensions improves.

**Figure 4: Long-term trajectory prediction** Figure 4 illustrates trajectory prediction for one of the joint dimensions which is computed by resorting to symplectic Euler numerical integration using the different GP models. The different GP models are the same as in Figure 2b using 1000 training points. Due to measurement noise in the initial state as well as numerical integration and prediction errors the GPs diverge from the true trajectory over time. As a consequence of Assumption 1 and 2, one can guarantee that trajectory predictions computed with (6) converge onto the surface equation $c(q) = 0$ by adjusting $b$ inside the S-GP model through Baumgarte stabilization [47] as detailed in Section 4 of the supplementary material. In turn, for the GP model's using Baumgarte stabilization the error in the position-level surface constraints converges to the zero.

## 4    Conclusion

In this paper, we introduce a modeling framework that combines differentiable Lagrangian dynamics with data-driven modeling. This framework incorporates structural prior knowledge in form of kinematic, inertia and implicit constraint equations to improve the data efficiency of a GP model. Further, we illustrate that the hyper-parameters of the GP and parameters of an analytical model can be estimated jointly using automatic differentiation. The framework requires that we have access to an accurate kinematics description of the system as well as that its bodies are sufficiently rigid. Notably, we illustrate that the dynamics of high-dimensional robot arms whose end-effector is subject to forces can be estimated using multi-output GPs.

Particularly in the combination with GP regression, the proposed framework opens up exciting research directions in the field of robotics modeling. So far, by letting the data-driven model approximate forces inside an analytical model, the data efficiency of the model is increased as we do not need to learn inertia and constraint functions from scratch. Yet, as we emphasized in the introduction, structured modeling can also improve a model's data efficiency by decreasing the target function's dimensionality. In particular, one can use the robot's differential kinematics to approximate with a GP the end-effector forces $\{F_z, F_I\}$ in the Cartesian space.

For the sake of illustration, we chose a SE kernel. In practice, such a kernel might not suffice as it assumes that the process is stationary and smooth. Both of these assumptions will likely not be met in practice. Instead, one can parameterize kernels using NNs [48, 49]. Further, rather than assigning all states as inputs to the kernel force prior, one can use additional prior model knowledge and use only physically relevant states as kernel inputs.

**Acknowledgments**

We thank Vincent Berenz for his valuable advice and the many insightful discussions. This work has been supported in part by the Max Planck Society and in part by the Cyber Valley initiative. The authors thank the International Max Planck Research School for Intelligent Systems (IMPRS-IS) for supporting A. René Geist.

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
