# OpenReview forum: "Using Physics Knowledge for Learning Rigid-body Forward Dynamics with Gaussian Process Force Priors"
_robot-learning.org/CoRL/2021/Conference — CoRL2021 Poster_

### Official Review · Reviewer_ZmdF · 2021-07-09

**Originality:** Very Good
**Technical Quality:** Good
**Clarity Of Presentation:** Fair
**Impact:** 4

**Recommendation:**

Weak Accept: I recommend accepting the paper, but will not argue for my recommendation if the majority of other reviewers have a different opinion.

**Summary:**

The paper proposes a method to learn forward dynamics while explicitly enforcing kinematic, inertia and implicit constraint equations. The constraint matrix, inertia matrix and known forces are explicitly estimated as the mean of the Gaussian Process (GP) Regression, while the variance of the GP captures uncertainties. The model parameters are trained using auto-differentiation. Experiment results in simulation show that, training models with structured knowledge converges faster and leads to smaller prediction errors.

**Issues:**

See "Strengths And Weaknesses" above.

**Reviewer Expertise:**

Poor: Limited knowledge of the area

**Strengths And Weaknesses:**

Strengths:
- The paper proposes a novel method to combine analytical and data-driven dynamics modeling. Despite its complexity, the model can be trained using auto-differentiation with fast GPU acceleration, which is promising.

Weaknesses:
- Lack of sufficient comparisons with existing baseline:
The author only compared their approach with standard GP and a standard feed-forward neural network. It would be nice to see the comparison with PETS [1], which uses an ensemble of probabilistic models to capture uncertainty. Reading from Fig.2 and 3 in the appendix, it seems like a major benefit of GP is the modeling of uncertainty and lower prediction errors, which is achievable by PETS.

- Organization:
The authors should better organize the main paper and appendices. Currently, the main paper focuses excessively heavy on problem setup and preliminary information, while a lot of interesting results and analyses remain in the appendix. My suggestions is as follows:
* Trim down section 1.2 and focus more on intuitions. What does $Q$, $Q_z$, $Q_I$ mean? Some examples could be helpful here ($Q_I, Q_Z$ in Fig.1 isn't exactly clear).
* Trim down the related works section. Right now it's overly detailed.
* Trim down section 2.1, or move it to Appendix.
* Move Fig.2/Fig.3 from the Appendix to the main paper, as well as related analysis. For example, why does the analytical model have significantly larger errors in dimension 4, 5, 6 only? Why does GP have significantly higher uncertainty in dimension 2 and 4 compared to other dimensions? It would be nice to investigate this in greater details.
* Move Section 5 in Appendix to the main paper, and add more results on long-term prediction accuracy.

References:
[1] Chua, Kurtland, et al. "Deep reinforcement learning in a handful of trials using probabilistic dynamics models." arXiv preprint arXiv:1805.12114 (2018).

**Summary Of Recommendation:**

[Update on Aug.30]

Update: I have updated the recommendation based on the author's responses. The paper's clarity is also improved in the revision. I have NOT updated the "clarity of representation" score yet.

[Original Comment]

The paper presents a novel way to model forward dynamics by combining both physics knowledge and data-driven regression. The main reason for the current recommendation is the organization: the main paper contains an excessive amount of details and preliminary information, while the results and analysis remain concise. I would love to re-evaluate the paper after revision.

---

> ### Author Response · Authors · 2021-08-30
> **Author Response (Part 1/2)**
>
> > Lack of sufficient comparisons with existing baseline: The author only compared their approach with standard GP and a standard feed-forward neural network. It would be nice to see the comparison with PETS [1], which uses an ensemble of probabilistic models to capture uncertainty. Reading from Fig. 2 and 3 in the appendix, it seems like a major benefit of GP is the modeling of uncertainty and lower prediction errors, which is achievable by PETS.
>
> We thank the reviewer for pointing us to PETS [1] as an alternative approach for probabilistic data-driven modeling of dynamics functions. **We added [1] to the introduction as we believe that similar to the GP based model-based RL framework PILCO, the results in [1] similarly indicate that probabilistic models may lead to better performance of model-based control algorithms.**
>
> As our paper focuses on the conscious combination of analytical mechanics with data-driven modeling, we tried to keep the comparison clear and simple. That is, comparing a GP with an SE kernel to the exact same GP with physics knowledge built into it, allows one to clearly deduce that the physics knowledge improved the model’s sample efficiency.
>
> The comparison to an analytical rigid-body model that is missing the dissipative force, also seems sensible as it highlights that a physics model’s performance is sensitive to unmodeled friction phenomena.
>
> **The comparison of the GP models to a NN could already be subject to debate, as the prediction accuracy of NN depends as strongly on its architecture as the prediction accuracy of a GP depends on the chosen Kernel function.** In turn, comparing a NN dynamics modeling approach such as PETS [1] to a structured GP with a comparably simple SE Kernel would not suffice. To get a sensible overview on how the state of the art in deep dynamics modeling compares to structured GP modeling, one would need to compare a careful selection of different “advanced” GP models with different deep dynamics modelling approaches. Yet, this, we believe, would result in a different type of paper as the one we intended to write. The focus of our submission lies on a discussion of implicit rigid-body mechanics and how a framework resulting from this discussion can be efficiently implemented to allow learning of high-dimensional dynamics functions.
>
> > Trim down section 1.2 and focus more on intuitions. What does Q, Q_z, Q_I mean? Some examples could be helpful here (Fig.1 isn't exactly clear).
>
> We added some intuition on the forces $Q$, $Q_z$, and $Q_I$. In particular, we write:
>
> *“[...] with the analytical model parameters $\theta_A$, symmetric and positive definite inertia matrix $M(q;\theta_A)$, torques acting on the unconstrained system $Q(x;\theta_A)$, torques $Q_z(x;\theta_A)$ caused by Cartesian end-effector friction forces $F_z$, and torques $Q_I(x;\theta_A)$ that are caused by Cartesian surface normal forces $F_I$ also acting on the end-effector. That is, if $Q$ presses the robot's end-effector onto a surface, the surface applies as a reaction the force $F_I$ that always remains normal to the surface. The transformation of $F_I$ into generalized coordinates yields the torques $Q_I$. The force $F_I$ often causes dissipative forces between the end-effector and the surface. These dissipative forces $Q_z$ lie always tangential to the surface. The torques acting in the unconstrained system $Q(x;\theta_A)=Q_D+Q_u+Q_G+Q_C$ consist of bias forces $Q_C$, gravitational forces $Q_G$, dissipative forces inside the joints $Q_D$, and actuation forces $Q_u$. Often in literature $Q_u+Q_D$ is denoted by $\tau$ being referred to as joint torques which are the torques that are measurable in a robot's joints.”*
>
> **We also added a table summarizing all forces to the supplementary material.**
>
> > Trim down the related works section. Right now it's overly detailed.
>
> We trimmed down the related works section.
>
> > Trim down section 2.1, or move it to Appendix.
>
> As suggested, we moved Section 2.1 to the Appendix to clear up some space for providing more intuition on rigid-body mechanics (Q, Q_z, Q_I ) as well as expanding on the discussion of the results.
>
> > Move Fig.2/Fig.3 from the Appendix to the main paper, as well as related analysis.
>
> We moved a smaller combined version of Figure 2 and Figure 3 to the main paper.
>
> > For example, why does the analytical model have significantly larger errors in dimension 4, 5, 6 only?
>
> We added the following sentence to the current submission that hopefully answers your question:
>
> *“The analytical baseline model does not contain a function describing the surface friction which in turn causes the model to yield large prediction errors in the robot arm's joints that are close to the end-effector.”*
>
>  So, the last 3 dimensions of the generalized accelerations (4,5,6) are the dimensions that are most affected by the unmodelled Cartesian friction forces when mapped to the generalized space.

---

> > ### Author Response · Authors · 2021-08-30
> > **Author Response (2/2)**
> >
> > > Why does GP have significantly higher uncertainty in dimension 2 and 4 compared to other dimensions? It would be nice to investigate this in greater details.
> >
> > Note that the measurement noise for all dimensions is the same. The conditional variance of the posterior GP distribution as depicted in Figure 4 in the supplementary material does not vary significantly between the different output dimensions. **As the y-axis scaling in Figure 4 of the supplementary material is not the same it gives the impression that the posterior variance varies significantly between the GP model’s outputs.** We added a sentence to the figure description that puts the reader’s attention to the differing y-axis scaling.
> >
> > > Move Section 5 in Appendix to the main paper, and add more results on long-term prediction accuracy.
> >
> > **We added an additional plot as well as a short discussion on the long-term prediction accuracy of the proposed structured model with and without Baumgarte stabilization.**

---

> > > ### Comment · Reviewer_ZmdF · 2021-08-30
> > > **Response to Authors**
> > >
> > > Thanks for the detailed response and the paper revision. I believe the authors have done an extensive job revising the paper, and the revision addresses most of my concerns. I have updated my rating accordingly.

---

### Official Review · Reviewer_HDXR · 2021-07-20

**Originality:** Very Good
**Technical Quality:** Very Good
**Clarity Of Presentation:** Very Good
**Impact:** 4

**Recommendation:**

Strong Accept: I recommend accepting the paper and will argue for my recommendation even if other reviewers hold a different opinion.

**Summary:**

The paper proposes a grey box model of rigid body dynamics (RBD) by combining a white box RBD model with a Gaussian process over external forces.
This model is evaluated in simulation of a contact-heavy manipulatior task.

**Issues:**

See above.

**Reviewer Expertise:**

Very good: Comprehensive knowledge of the area

**Strengths And Weaknesses:**

I thought this was a beautiful paper and was the best on my stack.
The idea of greybox modelling via the external forces is original and makes sense.
The inclusion of constrained dynamics is also a nice touch compared to previous data-driven RBD work..

The literature review was nice and informative. The only missing work I could think of is [A].

The technical section was very readable.

The weakness is in the experiments. For model learning, the real world is often very different to simulators (especially w.r.t. contract forces) so its a shame that we cannot see how the model scales to the real world. Do the authors have access to a real manipulator?

A second issue is with the scalability of GPs. I imagine sparse GPs slot into this framework quite easily, but would be good to see how the approximation effects the performance.

A neural network baseline of [20 30] is too small, especially has MBRL dynamics models are easily in the size of [200 200 200 200].

A final weakness is the title. The current title is far too general and doesn't distinguish this work from prior work. The title should include rigid body dynamics, Gaussian processes and their use as modelling external forces.

Line 339: isn't a quality of the SE kernel that it perfoms ARD?

Minor formatting suggestions:
- consider using boldface for matrices and vectors
- JMLR capitalization rules hold only for names of people and places
- it looks like you use \intercal and \top as your transpose symbol.


[A] A Bayesian Approach to Nonlinear Parameter Identification for Rigid Body Dynamics RSS Ting et al. 2006

**Summary Of Recommendation:**

Well executed paper on gray box model learning with GPs via external forces. A real robot experiment would be really good to properly evaluate the model learning.

---

> ### Author Response · Authors · 2021-08-30
> **Author Response**
>
> > The literature review was nice and informative. The only missing work I could think of is [A].
>
> Thank you for pointing us to [A]. We added [A] to the literature discussion.
>
> > The weakness is in the experiments. For model learning, the real world is often very different to simulators (especially w.r.t. contract forces) so its a shame that we cannot see how the model scales to the real world. Do the authors have access to a real manipulator?
>
> We agree that it would be nice to test the proposed framework on a real robot. We currently plan follow-up work that builds on the proposed structured GP framework to learn the dynamics of a real robot and which in particular delves more in the design of GP force priors for learning physical friction functions.
>
> > A second issue is with the scalability of GPs. I imagine sparse GPs slot into this framework quite easily, but would be good to see how the approximation effects the performance.
>
> The combination of sparse GPs with our framework is indeed an interesting direction to consider for follow-up work. For the current submission, considering the page limit of this submission, we are not sure if the analysis of approximation effects of sparse GP regression inside our structured modeling framework can be done thoroughly enough such that we arrive at a scientifically insightful conclusion.
>
> > A neural network baseline of [20 30] is too small, especially has MBRL dynamics models are easily in the size of [200 200 200 200].
>
> We agree that in literature on MBRL, the NN is considerably larger than the one we used in our results section. Yet, as the experiments shall show the limits of pure data-driven modelling compared to our proposed framework for structured modeling, we purposefully chose a data set that is considerably smaller than the data sets that one would encounter in MBRL literature. As pointed out, e.g. in [1], Neural Networks tend to overfit on small datasets, leading to bad prediction results on the test data set. This problem is aggravated by increasing the neural network size. While we tried to level the playing field in the results section by training the NN on ten times more data compared to our structured GP, we nonetheless found that the relatively small neural network had a lower test prediction error than larger networks. Nevertheless, we added to the appendix prediction results of a neural network of size of size [80] as well as [200 200 200 200].
>
> [1] Chua, Kurtland, et al. "Deep reinforcement learning in a handful of trials using probabilistic dynamics models." arXiv preprint arXiv:1805.12114 (2018).
>
> > A final weakness is the title. The current title is far too general and doesn't distinguish this work from prior work. The title should include rigid body dynamics, Gaussian processes and their use as modelling external forces.
>
> We changed the title to “Using Model Knowledge for Learning Rigid-Body Forward Dynamics with Gaussian processes”. Note that we do not mention external forces explicitly in the title, as the paper focuses in equal parts on the discussion of what qualifies as model knowledge as well as subsequently modeling external forces with GPs.
>
> > Line 339: isn't a quality of the SE kernel that it perfoms ARD?
>
> It is a nice quality of the SE kernel that it performs automatic relevance determination. However, despite this advantageous property, the SE kernel assumes that the process being modeled is infinite times differentiable which for most physical functions is not the case. This is why we point out in the conclusion of our submission, that future work could focus on designing GP kernels  for learning specific type of force functions. Another often mentioned reason for using the SE kernel is that it is a universal kernel, that is, it is capable of learning any continuous function given enough data, under certain conditions. We did not mention in our submission that SE kernels are universal, as force functions are often not continuous and structured models are used in scenarios where the amount of data is limited.
>
> > Minor formatting suggestions:
> consider using boldface for matrices and vectors
>
> We respectfully refrain from using boldface for matrices as in the revised manuscript almost every symbol denotes a vector or matrix. The only exceptions are the dimensions of the respective vectors or matrices.
>
> > JMLR capitalization rules hold only for names of people and places
>
> We are unfortunately not sure what we capitalized wrongly, **could you please point us to the type of words we may (de-)capitalize?**
>
> > it looks like you use \intercal and \top as your transpose symbol.
>
> Thank you for pointing this out. We now only use \top as transpose symbol.

---

### Official Review · Reviewer_Ahf1 · 2021-07-22

**Originality:** Fair
**Technical Quality:** Very Good
**Clarity Of Presentation:** Good
**Impact:** 3

**Recommendation:**

Weak Accept: I recommend accepting the paper, but will not argue for my recommendation if the majority of other reviewers have a different opinion.

**Summary:**

In this paper, the authors propose a structured learning approach to model the forward dynamics of a robot with implicitly constrained dynamics. For this purpose, they put a Gaussian Process prior on the unknown forces of the system. The basic structure of the equations of motion and the constrained dynamics are included in the Gaussian Process model by exploiting that GPs are closed under linear transformations. The hyperparameters are optimized by GPU accelerated computation. A simulation with a 7-dof robot shows the effectiveness of the proposed approach.

**Issues:**

Major:
- The problem setting is very sloppy. What means “sufficiently rigid”? What is an “accurate representation? Why “NOISY input point”? $q_k$, etc. are not defined at this point. What is an “informative data set”? “EOM” not defined here….
I would suggest to clarify in particular Assumption 2. The current form is not acceptable. Just consider rigid body motion.
- The general motivation to put the prior on the forces and not on the acceleration as in [35] in not sufficiently discussed. Furthermore, in line 152 “for some applications it can be more practically […] as we propose in this work? Which applications? Why?

Minor:
- The motivation for the usage of data-driven model is given by (stochastic) friction, backlash, etc. Yet, the simulation contains a very simple friction model (viscous friction). The results would be more convincing if you include more “advanced” types of friction in the system.
- Line 16/17: I would rather say any model-based control approach (includes standard control approaches such as computed torque for example)
- Line 26 and more: The term “analytical model” is not well defined. Do you mean parametric models? I think a GP is a (probabilistic) analytical model as well.
- Line 85: “$Q_u+Q_D$ […] being the joint torques that are measurable in the robot joints” Why is the gravitational force not measurable in the joints?
-Line 277: It’s a simulation and not an experiment.


**Reviewer Expertise:**

Very good: Comprehensive knowledge of the area

**Strengths And Weaknesses:**

Strength
-	High quality simulation that is described in detail in the supplementary material

Weakness
-	Sloppy problem setting and motivation
- Approach is very similar to existing work [35]


**Summary Of Recommendation:**

The problem presented in this paper is of interest and relevance. However, the solution seems to be very similar to [35] by simply exploiting the fact the GPs are closed under linear transformation. I do not see a contribution in the GPU acceleration as the MLE (14) is standard and GPU optimization packages are quite common nowadays. Thus, I see the main contribution in the thorough simulation. Therefore, it's mostly incremental theoretical work but with a nice simulation.

Update: Score remains unchanged. See reviewer comment for more information.

---

> ### Author Response · Authors · 2021-08-30
> **Author Response**
>
> > The problem setting is very sloppy. What means “sufficiently rigid”? What is an “accurate representation? Why “NOISY input point”? , etc. are not defined at this point. What is an “informative data set”? “EOM” not defined here…. I would suggest to clarify in particular Assumption 2. The current form is not acceptable. Just consider rigid body motion.
>
> We thank the reviewer for the detailed comments with regards to the problem formulation. We changed the wording of Assumption 2 to:
> *“The bodies (aka links) of the mechanical system are rigid.”*.
>
> We removed the terms *“noisy”* and *“informative”* from Section 1.1. The term *“EOM”* is now properly introduced.
>
> > The general motivation to put the prior on the forces and not on the acceleration as in [35] in not sufficiently discussed. Furthermore, in line 152 “for some applications it can be more practically […] as we propose in this work? Which applications? Why?
>
> We now compare our method to a structured model similar to [35] using the by us developed computation pipeline in PyTorch. The sentence in line 152,
>
> *“Albeit, for some applications it can be more practical to not rely on joint torque measurements and instead approximate QI and Qz through a data-driven model using constraint knowledge as we propose in this work.”*,
>
> we removed from the current submission due to space constraints. **We added examples for what phenomena $Q_z$ and $Q_I$ may represent in a real robot.**
>
> > The motivation for the usage of data-driven model is given by (stochastic) friction, backlash, etc. Yet, the simulation contains a very simple friction model (viscous friction). The results would be more convincing if you include more “advanced” types of friction in the system.
>
> Our paper proposes a conceptual as well as computational framework that details the combination of GP regression with rigid-body dynamics. Building on this framework, we plan follow-up work that explores the design of kernel force priors for learning structured dynamics that are influenced by differing types of friction and backlash.
>
> > Line 16/17: I would rather say any model-based control approach (includes standard control approaches such as computed torque for example)
>
> We use now in line 16/17 the wording  “model-based control approaches”.
>
> > Line 26 and more: The term “analytical model” is not well defined. Do you mean parametric models? I think a GP is a (probabilistic) analytical model as well.
>
> If the reader interprets “analytical” as “relating to or using analysis or logical reasoning”, this could indeed lead to confusion. We replaced the term “analytical model” with the term “physics model” or alternatively use the wording “analytical mechanics”.
>
> > Line 85: “[…] being the joint torques that are measurable in the robot joints” Why is the gravitational force not measurable in the joints?
>
> Let's assume that the robot's actuators are electric motors that are directly placed inside the robot arm's joints. Then torques are typically measured through the torsion of a (motor) shaft (of neglectable mass) due to two opposing torques being applied to the shaft’s ends. In the simplest case, the first end of the shaft lies embedded inside a motor while the other end of the shaft is attached to a body of the robot arm. On the second end of the shaft, torques are applied by either external forces (such as the gravitational force $Q_G$), inertia related torques $M\ddot q - Q_C$, end-effector related forces $Q_I$ and $Q_z$, or by dissipative forces inside the bearings of the joint $Q_D$. The only way to apply torques on the other end of the motor shaft is by the joint torques $\tau$, which can be motor control torques as well as dissipative torques inside the motor’s bearings holding the shaft. If the joint torques are zero, an external torque that is being applied to a robot arm’s body would simply rotate the body together with the shaft without twisting this shaft. Subsequently, a joint torque sensor measures the effect of the joint torques onto a shaft that is being attached to a rigid-body rotating around the joint. If no external forces act onto the robot arm except for $Q_G$, the joint torques change the robot arm’s motion according to the EOM $\tau = M \ddot{q} - Q_C - Q_G$. If we have an accurate model for $M \ddot{q} - Q_C$, we could estimate $Q_G$ using joint torque measurements.
> The above explanation reflects our understanding of measuring joint torques inside of robot arms. If the reviewer has additional insight on this topic, we would be thankful to receive additional remarks.
>
> > Line 277: It’s a simulation and not an experiment.
>
> We unfortunately missed to change the wording "experiment" to "simulation" in the current revised version. However, we will change the wording accordingly in a subsequent revised version.

---

> > ### Comment · Reviewer_Ahf1 · 2021-08-31
> > **Reviewer response**
> >
> > Thanks to the authors for the updated paper. However, I still have some comments:
> >
> > > We removed the terms “noisy” and “informative” from Section 1.1. The term “EOM” is now properly introduced.
> >
> > In line 63, you are still using the term "noisy". If that's not a mistake, please explain why.
> >
> > >Our paper proposes a conceptual as well as computational framework that details the combination of GP regression with rigid-body dynamics.
> >
> > I understand conceptual work but what's the benefit of the idea if you can't convince the reader that your approach is needed for some settings/applications? I strongly suggest to use a more complex simulation example. In addition, the concept is still very similar to [35] such that my score remains the same even though I appreciate the work of the authors.

---

### Official Review · Reviewer_2B5H · 2021-07-22

**Originality:** Fair
**Technical Quality:** Very Good
**Clarity Of Presentation:** Very Good
**Impact:** 3

**Recommendation:**

Weak Accept: I recommend accepting the paper, but will not argue for my recommendation if the majority of other reviewers have a different opinion.

**Summary:**

This paper presents a method for embedding prior knowledge on the dynamics of a robot (e.g. in the form of physical constraints that the system has to satisfy, the inertia matrix, etc.) inside a dynamics-learning framework in order to help the model generalize better. This is done by modeling the robot dynamics as a Gaussian process, and incorporating the a-priori known dynamics information in the GP's mean function. The method is benchmarked on a 7-DOF arm following a planar end-effector trajectory and the dynamics models learned using the approach are shown to generalize better than baseline approaches.

**Issues:**

- I would be more convinced of the utility of the method if there was a baseline comparison with [35], with the training procedure changed to use GPyTorch-based inference.
- I would also be more convinced if the authors could provide justification on where the a-priori information on the unknown parameters might come from.
- It would help to expand the discussion of the results and clarify the usage of \theta_A, as described above.

**Reviewer Expertise:**

Very good: Comprehensive knowledge of the area

**Strengths And Weaknesses:**

Strengths
- The paper is very clearly written, providing an extensive motivation of the problem studied and an extensive overview of prior work.
- The paper is technically solid, though it is admittedly easy to get lost in the notation at times (a table summarizing all the different forces would be helpful for the reader).

Weaknesses
- Overall, the contribution appears to be a somewhat minimal modification to reference [35] in the paper. From my understanding, the differences are twofold: 1) swapping out modeling the unknown accelerations with modeling the unknown forces and 2) implementing the GP training in GPyTorch. The paper claims on line 192 that the first change can improve sample efficiency, yet there is no baseline comparison in the results with a method that learns the accelerations directly (i.e. a variant of [35] which is trained using GPyTorch), which leaves me uncertain of the relative improvement of this paper's method over [35].
- The use of \theta_A is rather confusing, as it is used to refer to different subsets of the mass matrix/forces and to both known and unknown parameters. In the discussion near line 80, \theta_A refers to any model parameters, including the mass matrix and all forces. In line 231, it is used to denote possibly erroneous parameters of the mass matrix M, but in line 243 it is claimed that M is known. Later in the results section (line 300), it seems to be instead used to denote components of Q_G, Q_C, and Q_u. It would help clarify things if \theta_A were explicitly described for each example involving the 7-DOF arm in the results.
- While the primary benefit of modeling the forces instead of the accelerations is to take advantage of prior knowledge in known components of the forces/the inertia matrix, the paper doesn't sufficiently motivate why we might have prior knowledge for some of the discussed components. The mass matrix M and A, b (arising from the constraint manifold) are motivated in the paragraphs, but it's not clear why we might have prior knowledge on Q_C or Q_D, for instance. More discussion on why it might be plausible to have prior knowledge on each analytical dynamics parameter would be useful.
- The results are a bit confusing:
---- In Figure 2b, why are there multiple curves with the same color? Are these just the loss curves for different training runs?
---- There is a baseline comparison to a neural network discussed in the text, which based on those numbers seems to have similar performance as the proposed methods, but it does not appear in Figure 2; why is this? Also, from the supplementary, it seems like the NN is trained using more points than the GP-based methods; using the same number of points would help ease comparison. More discussion comparing the NN to the proposed approach would help.
---- Q is modeled as Q_u + Q_G + Q_C in this example; why are dissipative forces Q_D not considered?

**Summary Of Recommendation:**

While the paper is overall well-written and technically proficient, I am not fully convinced of the method's contribution compared to prior work [35] or the plausibility of having prior information on certain dynamics parameters. A revision along the lines of what follows would help improve my confidence in the paper.

---

> ### Author Response · Authors · 2021-08-29
> **Author Response (Part 1/2)**
>
> > The paper is technically solid, though it is admittedly easy to get lost in the notation at times (a table summarizing all the different forces would be helpful for the reader).
>
> We agree that the chosen notation for the robot’s dynamics equations deviate from what could be considered a standard notation in literature on robot dynamics as found e.g. in [1] - [3]. We chose this particular notation to emphasize that we distinguish analytical mechanics knowledge in the lines of inertia-related functions (denoted by M) and force-related functions (denoted by Q). **We added a table summarizing all dynamic terms to the appendix of the current submission.**
>
> [1] Bruno Siciliano, Lorenzo Sciavicco, Luigi Villani, and Giuseppe Oriolo. Robotics: modelling, planning and control. Springer Science & Business Media, 2010.
> [2] Roy Featherstone. Rigid body dynamics algorithms. Springer, 2008.
> [3] Werner Schiehlen and Peter Eberhard. Applied dynamics, volume 57. Springer, 2014.
>
> > Overall, the contribution appears to be a somewhat minimal modification to reference [35] in the paper. From my understanding, the differences are twofold: 1) swapping out modeling the unknown accelerations with modeling the unknown forces and 2) implementing the GP training in GPyTorch. The paper claims on line 192 that the first change can improve sample efficiency, yet there is no baseline comparison in the results with a method that learns the accelerations directly (i.e. a variant of [35] which is trained using GPyTorch), which leaves me uncertain of the relative improvement of this paper's method over [35].
>
>  Using our framework, we added the prediction results of a structured GP model in the lines of [35] to Figure 2. Figure 2 now shows the relative improvement of our proposed framework compared to a structured model that places a GP prior onto the system’s accelerations.
>
> > The use of \theta_A is rather confusing, as it is used to refer to different subsets of the mass matrix/forces and to both known and unknown parameters. In the discussion near line 80, \theta_A refers to any model parameters, including the mass matrix and all forces. In line 231, it is used to denote possibly erroneous parameters of the mass matrix M, but in line 243 it is claimed that M is known. Later in the results section (line 300), it seems to be instead used to denote components of Q_G, Q_C, and Q_u. It would help clarify things if \theta_A were explicitly described for each example involving the 7-DOF arm in the results.
>
> We thank the reviewer for this observation and agree that especially line 80 and line 231 may confuse a potential reader. $\theta_A$ shall denote all parameters of a given rigid-body dynamics model as pointed out in line 80. To make this point more clear, we added an additional sentence to the submission close to line 80, reading:
>
> *“The analytical model parameters $\theta_A$ can be divided into kinematic parameters (e.g. the length between adjacent joints) and inertia parameters (\eg CoG positions, masses, and inertias). The model parameters $\theta_A$ often deviate from the real physical values of the system.”*
>
> We further adjusted the mentioned sentence in line 231 to:
> *“Yet, for FD it is critical that we mitigate errors in the inertial parameters **being part of $\theta_A$** as all forces in (4) are being multiplied by $M^{-1}(\theta_A)$.”*
>
> While we assume that $M$ is known in line 243, we use our definition of known as given in line 234 as:
>
> *“The term known is defined in the following as the existence of an optimal $\theta_A$, $\theta_A^\star$, such that the error between a function and its analytical description approaches zero.”*
>
> To improve the clarity of this definition, we slightly altered the definition in the current submission in line 234 to:
>
> *“The term known is defined in the following as the existence of an optimal $\theta_A$, $\theta_A^\star$, such that the error between a **physical **function and its analytical description **in terms of $\theta_A$** approaches zero.”*
>
> To improve clarity, we altered line 300 to:
> *“In this figure, the term "S-GP" refers to the model in (6) **with $Q_K=0$** while the term ``S-GP + analytical mean'' assumes an analytical model $Q_K=Q_G+Q_C+Q_u$ as the GP's prior mean function. **For both S-GP models we assume that accurate analytical parameters are given, that is** $\theta_A=\theta_A^\star$”*

---

> > ### Author Response · Authors · 2021-08-29
> > **Author Response (Part 2/2)**
> >
> > > While the primary benefit of modeling the forces instead of the accelerations is to take advantage of prior knowledge in known components of the forces/the inertia matrix, the paper doesn't sufficiently motivate why we might have prior knowledge for some of the discussed components. The mass matrix M and A, b (arising from the constraint manifold) are motivated in the paragraphs, but it's not clear why we might have prior knowledge on Q_C or Q_D, for instance. More discussion on why it might be plausible to have prior knowledge on each analytical dynamics parameter would be useful.
> >
> > We thank the reviewer for these helpful remarks. Around line 300, **we added the following sentence to emphasize why prior knowledge on $Q_C$ as well as $Q_G$ is plausible:**
> >
> > *“We assume $Q_G+Q_C$ as known as these analytical functions are being derived solely in terms of known kinematics and inertia functions, as well as the gravitational acceleration constant [17].”*
> >
> >  In [1], the exact analytical equations of $Q_C$ and $Q_G$ are denoted in terms of kinematic functions as well as $M$ for the case of Lagrangian dynamics that are being subject to implicit constraints. **We do not assume prior knowledge on the dissipative force $Q_D$ or $Q_z$.** As these forces being dissipative, they arise from friction or damping phenomena that are often difficult to model a priori.
> >
> > [1] A. R. Geist and S. Trimpe. Structured learning of rigid-body dynamics: A survey and unified view from a robotics perspective. GAMM Mitteilungen, 44(2):34, 2021.
> >
> > > --- In Figure 2b, why are there multiple curves with the same color? Are these just the loss curves for different training runs?
> >
> > The plots were indeed unnecessarily confusing. The y-axis in Figure 2b showed the mean prediction error on a test data set for **each** output dimension (joint) of the respective dynamics model. In the current submission, Figure 2b now shows the mean prediction error over all output dimensions of the structured dynamics model. The x-axis is kept the same showing an increasing number of training points that are used to train the respective dynamics model.
> >
> > > --- There is a baseline comparison to a neural network discussed in the text, which based on those numbers seems to have similar performance as the proposed methods, but it does not appear in Figure 2; why is this?
> >
> > The performance of the neural network (NN) is only comparable with the GP models using 10 times the maximum number of data points. **The prediction error of the NN for using 100 to 1000 training points is considerably larger than the errors of the GP models.** If we would plot the large prediction error of the NN in Figure 2b, one could barely see a difference in the GP's prediction errors.
> >
> > > Also, from the supplementary, it seems like the NN is trained using more points than the GP-based methods; using the same number of points would help ease comparison. More discussion comparing the NN to the proposed approach would help.
> >
> > Already at ten times the maximum number of data points used for estimating the GP hyperparameters, the analytical model and NN have a larger prediction error than the GP models. As reducing the number of data points for training the analytical model and NN is likely to  increase the error of these models, we do not see the necessity to add these models to Figure 2.
> >
> > > --- Q is modeled as Q_u + Q_G + Q_C in this example; why are dissipative forces Q_D not considered?
> >
> > We split $Q$ in a known part $Q_K$ and an unknown part that shall be modeled using a GP kernel prior. As already discussed above, the bias force $Q_C$ and the gravitational force $Q_G$ can be assumed to be known, as these terms can be derived analytically by solely using kinematics, inertia functions, and the gravitational acceleration constant. In the case of electric motors, the control force $Q_u$ can be closely approximated using the supplied motor current and motor-torque constant. **Terms that are unknown, that is, cannot be solely described in terms of a priori derived analytical quantities, are the dissipative forces $Q_D$ and the end-effector force $Q_z$.** These unknown forces are being approximated by the Gaussian process which is then being transformed by the known inertia matrix to yield the system’s acceleration caused by $Q_D + Q_z$.
> >
> > **In short, $Q_D + Q_z$ being difficult to model analytically a priori is being approximated by the GP using data.**

---

> > > ### Comment · Reviewer_2B5H · 2021-09-02
> > > **Reviewer Response**
> > >
> > > Thank you for addressing the points in my review. The comparison to prior work is useful to show the benefit of the method when less data is available. While I don't believe there is a lot of novelty in modeling forces vs. accelerations, it does produce better results, which is noteworthy. I will change my rating to "Weak Accept".

---

### Meta-Review · Area_Chair_KGne · 2021-08-16

**Recommendation:** Accept (Poster)
**Confidence:** 4

**Metareview:**

This paper proposes the use of additional structure/model knowledge to improve the learning of forward dynamics models.

Quality:

(-) Reviewers 2B5H, Ahf1, and ZmdF have concerns about the contribution of the manuscript compared to existing literature. Adding an experimental comparison to past literature would significantly strengthen the paper and better demonstrate the contributions of the proposed approach.

Clarity:

(+) The paper is generally clear

(-) Reviewers Ahf1 and ZmdF noted that the problem setting should be improved.

(-) All the reviewers suggested several improvements in notation, presentation of the results, and even title. Please address these comments.

Originality:

(-) There have been several works in the literature exploring the use of structured priors and/or model priors to improve the learning of forward dynamics models.

(+) To my knowledge, the use of this specific parametrization is novel.

Significance:

(+) This topic is important and impactful

(-) As highlighted by Reviewers 2B5H and ZmdF, the manuscript does not compare against any of the existing literature. As such, it is impossible to evaluate the significance of the contribution.

---

> ### Author Response · Authors · 2021-08-29
> **Author Response (Part 1/2)**
>
>  We thank the Area Chair as well as the Reviewers for their time and careful evaluation of our manuscript. We have restructured the manuscript according to the reviewer's suggestions, and provide below detailed responses to all comments. We are confident that we have now addressed the open issues of the previous submission, and hope that the reviewers find the changes convincing.
>
> Further, we attached a **Latex diff file of the main manuscript to the supplementary material**, where all changes to the previous submission are being highlighted.
>
>
> > (-) Reviewers 2B5H, Ahf1, and ZmdF have concerns about the contribution of the manuscript compared to existing literature. Adding an experimental comparison to past literature would significantly strengthen the paper and better demonstrate the contributions of the proposed approach.
>
> As suggested by 2B5H and Ahf1, **we compare now our model to our implementation of [35]** (being reference [37] in the revised manuscript) using our computational framework. Here, we now compare the difference between placing a GP prior on the system’s unknown accelerations to placing a GP prior on the unknown forces. Further, **we incorporated the suggestions of Reviewer Zmdf with regards to restructuring the paper** and extending the discussion of the results.
>
> > (-) Reviewers Ahf1 and ZmdF noted that the problem setting should be improved.
>
> We revised the problem formulation using the reviewer's feedback.
>
> > (-) All the reviewers suggested several improvements in notation, presentation of the results, and even title. Please address these comments.
>
> We incorporated the suggestions on the notation and presentation into the manuscript. We changed the title of the submission to  “Using Model Knowledge for Learning Rigid-Body Forward Dynamics with Gaussian processes”.

---

> > ### Author Response · Authors · 2021-08-29
> > **Author Response (Part 2/2)**
> >
> > > (-) There have been several works in the literature exploring the use of structured priors and/or model priors to improve the learning of forward dynamics models.
> >
> > Certainly there exist prior works that explore the use of structured priors and/or model priors to improve the learning of forward dynamics models. Yet, these works differ strongly from our work both in terms of the underlying motivation as well as in terms of the proposed modeling framework. We also discussed these prior works in the introduction of our submission. As reviewer HDXR wrote with regards to our literature discussion *“The literature review was nice and informative.”* while reviewer 2B5H wrote, *“The paper is very clearly written, providing an extensive motivation of the problem studied and an extensive overview of prior work.”*
> >
> > **To the best of our knowledge, there exists no other work that details a framework for combining multi-output GP regression with rigid-body forward dynamics for high-dimensional robot arms.** To provide such a framework, we implemented a recursive rigid-body dynamics library in PyTorch as well as also designed from scratch a library for structured GP modeling/inference using PyTorch as well as GPyTorch’s mean and covariance function. As our submission provides a clearly structured code repository alongside the publication, we believe that from a practical point of view our work significantly eases future research on the combination of rigid-body dynamics with data-driven regression modeling.
> >
> > From a theoretical point of view, our work distinguishes itself from prior literature succinctly yet more subtly in the following points:
> >
> > - While data-driven dynamics modeling aims at reducing the error between a dynamics model and the true dynamics, **we emphasize that the actual motivation behind structured modeling is to improve the model’s sample efficiency and extrapolation capability**. Subsequently, we state around line 30, *“By use of physical prior knowledge, the data efficiency of a structured model is potentially improved by: i) Reducing the complexity of the target function; ii) Reducing the dimensionality of the target function; or iii) Building prior knowledge on physical properties of the target function into the data-driven model.”* **This division of the potential ways that structured modeling may improve the sample-efficiency of a data-driven model, we did not encounter in prior literature.** We believe this statement is of utmost significance for structured modeling as it emphasizes the potential advantage of using forces as target functions instead of accelerations. As we point out in the conclusion of our work, *“So far, by letting the data-driven model approximate the forces inside an analytical model, the data efficiency of the model has been increased as we do not need to learn the inertia and constraint functions from scratch. Yet, as we emphasized in the introduction, structured modeling can also improve a model’s data efficiency by decreasing the dimensionality of the target function”.* In short, our work points to the **numerous advantages of placing data-driven model priors on forces instead of accelerations as in [35],** and in turn hopefully inspires future research to explore the above mentioned different ways of improving the sample-efficiency of a structured model.
> >
> > - We point out that **Baumgarte stabilization** is not only a useful numerical technique for the simulation of rigid-body dynamics, but also **can straightforwardly enforce position-level constraint satisfaction of the proposed structured dynamics model**.
> >
> > - [35] showed that a structured GP allows to predict the system’s unconstrained acceleration by conditioning a GP joint distribution on measurements of the constrained system. We significantly extended this idea in our work, as **our framework allows the prediction of various mechanical quantities such as implicit constraint forces**. We believe that the possibility to use a data-driven regression model being trained on acceleration observations to also predict when the system is leaving the implicit constraint equation is useful for robotics, e.g. predicting the motion of quadrupedal robot.
> >
> > > (-) As highlighted by Reviewers 2B5H and ZmdF, the manuscript does not compare against any of the existing literature. As such, it is impossible to evaluate the significance of the contribution.
> >
> > Except of [35], we did not encounter prior work that combines GP regression with forward rigid-body dynamics models. We agree that a comparison to an extension of [35] is insightful and added such a comparison to the results section. **The comparison shows that for the robot arm dynamics placing a GP prior on the system’s acceleration rather than its forces results in a larger prediction error.**

---

> > > ### Comment · Area_Chair_KGne · 2021-09-09
> > > **After rebuttal**
> > >
> > > After rebuttal, there seems to be agreement from the reviewers that the work has been improved and is solid.
> > >
> > > Two final notes:
> > > - Regarding the claim that "there exists no other work that details a framework for combining multi-output GP regression with rigid-body forward dynamics" and that your work is substantially different from prior literature, I do not buy this argument and I would be very careful with this claim. For example, "Data-Efficient Control Policy Search using Residual Dynamics Learning" by Saveriano et al is proposing a comparable framework.
> > > - I would strongly suggest that you reconsider the suggestion by Reviewer HDXR to incorporate the term "acceleration" (or equivalent) in the title. The new title "Using Model Knowledge for Learning Rigid-Body Forward Dynamics with Gaussian processes" is still to generic and doesn't really reflect the contribution that you are providing.

---

### Decision · Program_Chairs · 2021-09-13

**Decision:**

Accept (Poster)

**Comment:**

This paper proposes the use of additional structure/model knowledge to improve the learning of forward dynamics models.

Quality:

(-) Reviewers 2B5H, Ahf1, and ZmdF have concerns about the contribution of the manuscript compared to existing literature. Adding an experimental comparison to past literature would significantly strengthen the paper and better demonstrate the contributions of the proposed approach.

Clarity:

(+) The paper is generally clear

(-) Reviewers Ahf1 and ZmdF noted that the problem setting should be improved.

(-) All the reviewers suggested several improvements in notation, presentation of the results, and even title. Please address these comments.

Originality:

(-) There have been several works in the literature exploring the use of structured priors and/or model priors to improve the learning of forward dynamics models.

(+) To my knowledge, the use of this specific parametrization is novel.

Significance:

(+) This topic is important and impactful

(-) As highlighted by Reviewers 2B5H and ZmdF, the manuscript does not compare against any of the existing literature. As such, it is impossible to evaluate the significance of the contribution.